# sRGB Real Noise Modeling via Noise-Aware Sampling with Normalizing Flows

**Dongjin Kim**[*1]**, Donggoo Jung**[*2]**, Sungyong Baik**[3]**, Tae Hyun Kim**[†1]
Dept. of Computer Science[1], Dept. of Artificial Intelligence[2], Dept. of Data Science[3]
Hanyang University
{dongjinkim, dgjung, dsybaik, taehyunkim}@hanyang.ac.kr

## Abstract

Noise poses a widespread challenge in signal processing, particularly when it comes to denoising images. Although convolutional neural networks (CNNs) have exhibited remarkable success in this field, they are predicated upon the belief that noise follows established distributions, which restricts their practicality when dealing with real-world noise. To overcome this limitation, several efforts have been taken to collect noisy image datasets from the real world. Generative methods, employing techniques such as generative adversarial networks (GANs) and normalizing flows (NFs), have emerged as a solution for generating realistic noisy images. Recent works model noise using camera metadata, however requiring metadata even for sampling phase. In contrast, in this work, we aim to estimate the underlying camera settings, enabling us to improve noise modeling and generate diverse noise distributions. To this end, we introduce a new NF framework that allows us to both classify noise based on camera settings and generate various noisy images. Through experimental results, our model demonstrates exceptional noise quality and leads in denoising performance on benchmark datasets.

## 1 Introduction

Noise is one of the most common and undesired artifacts in the field of signal processing. Therefore, denoising has been studied for many decades across many research areas, among which image denoising is a task to restore a clean image by removing noise from the input image. Recently, there have been dramatic improvements in image denoising through the development of convolutional neural networks (CNN) (Zhang et al., 2017; Anwar & Barnes, 2019; Chen et al., 2021) trained in a supevised manner. Despite their impressive achievements, these supervision-based algorithms have critical limitations in that they rely on a simple assumption that added noise follows known distributions, such as Gaussian and Poisson. Such an assumption hinders the supervised algorithms from generalizing to real-world noises that do not follow the well-known, simple distributions.

To enable supervised approaches in handling real-world noise, there have been several attempts to collect real-world datasets consisting of pairs of clean images and their corrupted versions with real noise (Plotz & Roth, 2017; Nam et al., 2016; Xu et al., 2018; Anaya & Barbu, 2018; Abdelhamed et al., 2018; 2020). Among them, one of the widely used datasets for training is the SIDD (Abdelhamed et al., 2018) dataset, which consists of real-world noisy images captured by various smartphone cameras, considering different ISO, shutter speed, and aperture settings. Nevertheless, obtaining such images requires a significant amount of time and resources, which restricts the availability of large-scale real noisy datasets.

To alleviate these constraints, generative methods have been proposed. Specifically, numerous generative approaches employ generative adversarial networks (GAN) or normalizing flows (NF) to learn the distribution of real noise and synthesize realistic noisy images (Yue et al., 2020; Zamir et al., 2020; Abdelhamed et al., 2019; Jang et al., 2021; Chang et al., 2020; Kousha et al., 2022).

In this study, we introduce a novel framework called NAFlow, short for **N**oise-**A**ware Normalizing **Flow**, designed to learn and effectively manage the diverse noise distributions originating from various camera settings.

---

[*] Equal contribution. [†] Corresponding author.

Unlike previous models such as NeCA (Fu et al., 2023) that use multiple different noise models to deal with different noise distributions, we use only a single unified model to learn such complex noise distributions. Additionally,

Table 1: Comparison between sRGB noise modeling networks.

| Key features | Flow-sRGB | NeCA | Ours |
|---|---|---|---|
| Sampling without metadata | ✗ | ✗ | ✓ |
| Single unified model | ✓ | ✗ | ✓ |
| Correlated noise generation | ✗ | ✓ | ✓ |

we present a **N**oise-**A**ware **S**ampling (**NAS**) algorithm, which empowers our noise generation model to exploit the Gaussian mixture model (GMM) derived from diverse noise distributions. Importantly, NAFlow mitigate the necessity for metadata (*e.g.,* ISO, camera model) associated with input noisy images for sampling. Furthermore, we recognize the importance of spatial noise correlation in sRGB space, and thus propose a multi-scale noise embedding technique to facilitate the synthesis of the spatially correlated noise. Through various experiments, NAFlow has demonstrated its outstanding noise quality. Additionally, when utilized for denoising tasks with generated noise, it has achieved a state-of-the-art level of performance on SIDD benchmark.

## 2  RELATED WORK

**Noisy Image Generation.** Due to the lack of real-noise datasets, there are several attempts to generate realistic noisy images. DANet (Yue et al., 2020) learns the joint distribution of the clean and noisy image pair to map the noisy image to the clean one, enabling simultaneous noise generation and elimination. C2N (Jang et al., 2021) is trained with unpaired clean and noisy images and generates synthetic noisy images with adversarial loss. CycleISP (Zamir et al., 2020) presents a CNN-based framework and exploits camera image pipelines on both the sRGB and RAW spaces. CA-NoiseGAN (Chang et al., 2020) encodes camera-dependent noise information using contrastive learning and generates realistic noise conditioned on that information. PNGAN (Cai et al., 2021) splits the noisy image generation problem into image-domain and noise-domain alignment problems and carries out pixel-level noise modeling. Moreover, there are some generative models based on conditional NF rather than GANs since NF can be trained easily and generate more diverse images (Lugmayr et al., 2020; Abdelhamed et al., 2019; Kingma & Dhariwal, 2018; Wang et al., 2022; Liang et al., 2021; Kousha et al., 2022). NoiseFlow (Abdelhamed et al., 2019) learns noise distribution of RAW images captured by smartphone cameras conditioned on the smartphone type and gain setting (*e.g.,* ISO), and Flow-sRGB (Kousha et al., 2022) extends the capability of NoiseFlow to tackle denoising in sRGB space. However, the existing noise modeling methods lack modeling spatially correlated noise. To address this issue, NeCA (Fu et al., 2023) explicitly learns the neighboring correlation of the noise within the dataset of SIDD (Abdelhamed et al., 2018). Although NeCA closes the distribution gap between synthetic and real noise in the sRGB space, it requires multiple different noise models to deal with different real-world noise.

In this work, we extend the capabilities of the NF-based framework as a generative classifier as well as the ability to generate a variety of samples, as discussed in previous studies (Mackowiak et al., 2021; Izmailov et al., 2020; Ardizzone et al., 2020). In particular, we employ GMM to learn multiple different real-world noise distributions according to ISO levels, camera models, which mainly affect the noise distribution in camera imaging pipeline (Ramanath et al., 2005; Gow et al., 2007). These learned distributions enable effective processing and generation of real input noisy images. For clarity, Tab. 1 illustrates the differences between recent noise modeling methods and our approach. Our objective is to sampling realistic noisy that considers noise correlation during the sampling phase without any need of metadata, while employing only a single unified model.

## 3  BACKGROUND

### 3.1  NOISE MODELING

In this subsection, we begin by explaining the process of image noise acquisition, and introduce several noise modeling methods. Noisy images can be represented as a combination of a clean image and noise using the following equation:

$$y = x + n, \tag{1}$$

Figure 1: Overview of proposed framework. (a) The training procedure of our NAFlow, which uses conditional normalizing flow network $f$ to map noisy images to Gaussian distributions on the latent space, where each Gaussian distribution corresponds to the camera configuration used for capturing images. (b) The pipeline of inference procedure NAS. We use the Gaussian mixture model to obtain a more accurate latent representation without metadata of a given input noisy images.

where $x$ and $y$ represent the clean and noisy images, respectively, and $n$ represents the added noise.

Real-world image noise $n$ arises from inherent constraints associated with camera sensors and is further introduced during the post-imaging process. This real-world noise—including photon noise, read noise, and quantization—occurs during the initial capture of a RAW-RGB image. Subsequently, a RAW-RGB image undergoes non-linear imaging processes (such as demosaicking, gamma correction, and tone mapping), transforming it from a scene-based RAW-RGB color space to a screen-based sRGB color space. Due to the application of multiple non-linear operations to both a clean image and a noise, modeling sRGB noise becomes a more challenging problem. To address this problem, we aim to transform the complex real-world noise distributions into several simple distributions to leverage the diverse properties of real-world noise by adopting normalizing flows.

## 3.2 NORMALIZING FLOW

Normalizing flows (NF) belong to a class of generative models that use invertible transformations to map a simple distribution, typically a Gaussian distribution, to a more complex target distribution. The invertible mapping allows NF to directly optimize negative log likelihood (NLL). Due to the advantageous combination of simplicity and effectiveness offered by NF (Abdelhamed et al., 2019; Kousha et al., 2022), we use NF for modeling noise distributions.

Composed of a sequence of invertible transformations, our NF aims to learn an invertible mapping $f : z \mapsto x$, where $x$ and $z$ are the original (image) and transformed data (latent), respectively. Specifically, the change of variables formula is employed to compute the probability density of the latent $z$ with respect to the original data $x$, as follows:

$$p(x) = p(z) \cdot |\mathrm{det} D f_\theta(x)|. \tag{2}$$

Where, $\mathrm{det} D f_\theta(x)$ represents the determinant of the Jacobian matrix of the transformation $D f_\theta(x)$, taking into account how the volume changes during the mapping. The training objective is the NLL loss of the observed data $x$, which is minimized by training the model parameters $\theta$.

## 4 PROPOSED METHOD

In this section, we start with the description of our NAFlow, a novel NF-based framework that allows for learning multiple distinctive Gaussian distributions within the latent space, thereby enabling the effective modeling of a diverse real noise distributions (Sec. 4.1). Next, we present our NAS algorithm, which uses the trained NF not only as a noise generator but also as a noise classifier during the inference phase (Sec. 4.2). In contrast to previous works (Kousha et al., 2022; Fu et al., 2023) that require the meta-data of target noise even during inference, we reduce the dependency through our NAS algorithm. Specifically, our NAS algorithm measures similarities between the input noise and several learned real-world noise distributions. And then, we synthesizes noise that follows a similar distribution to the input noise. Finally, we present a multi-scale noise embedding strategy to capture spatially correlated noise features within the sRGB input image. (Sec. 4.3).

(a) S6_00400     (b) G4_00400     (c) IP_01000     (d) N6_03200     (e) GP_06400

Figure 2: Noise visualization from diverse device settings (*e.g.,* smartphone model, ISO). Each noise has a unique distribution due to the distinct characteristics of each image signal processor.

## 4.1 LEARNING MULTIPLE DISTINCT NOISE DISTRIBUTIONS

In this work, we aim to generate realistic noisy images to solve the real-world denoising problem. To generate such synthetic noisy images, it is required to capture distributions of real noisy images as in (Yue et al., 2020; Jang et al., 2021; Fu et al., 2023). To do so, we learn the complex distribution of noisy images in the SIDD dataset (Abdelhamed et al., 2018), which consists of real noisy images with corresponding meta-information and clean counterparts. Note that the real-world noise distribution is unknown and varies depending on the camera model, sensor type, and camera settings (Ramanath et al., 2005; Gow et al., 2007). Furthermore, as shown in Fig. 2, we observe distinct noise correlation conditioning on each camera configuration (*e.g.,* smartphone model, ISO) in the SIDD. This observation motivates us to model multiple camera-configuration-specific normal distributions in latent space to handle various real-world noise.

Following the usual settings in the field of noise generation (Yue et al., 2020; Kousha et al., 2022; Fu et al., 2023), we assume that ground truth clean images are given, and we use the clean images as conditional information, where this conditional information allows the NF to learn a condition-specific noise distribution. Specifically, in our work, we use the conditional NF (Ardizzone et al., 2019) which takes a ground truth clean image $x$ for conditioning as:

$$z_c = f_\theta(y_c; x) \iff y_c = f_\theta^{-1}(z_c; x), \tag{3}$$

where $y_c$ denotes a real noisy image taken with camera configuration $c$, and our camera configuration $c$ consists of smartphone model and ISO setting (*e.g.,* iPhone7 with ISO 1600). The noisy image undergoes mapping to a latent variable $z_c$ through the invertible conditional normalizing flow $f$ with parameters $\theta$, and ground truth clean image $x$ is used for conditioning the NF.

Then, we learn the noise distributions using camera configuration-specific normal distributions through a single NF as follows:

$$z_c \sim \mathcal{N}(\mu_c, \Sigma_c), \tag{4}$$

where $\mu_c$, $\Sigma_c$ denotes the mean and covariance matrix of the specific normal distribution for the configuration $c$ and are trainable parameters in this work. Then, we can compute the conditional probability density function of $y_c$ given $x$ as:

$$p(y_c|x) = \mathcal{N}(z_c; \mu_c, \Sigma_c) \cdot |\det D f_\theta(y_c; x)|, \tag{5}$$

where $D f_\theta(y_c; x)$ denotes Jacobian of $f_\theta$. To handle multiple different distributions using a single NF, we train the NF by minimizing the negative log-likelihood (NLL) loss function as follows:

$$\begin{aligned}
\mathcal{L}_{NLL}(\theta) &= -\sum_{c=1}^{C} \log p(y_c|x) \\
&= -\sum_{c=1}^{C} (\log \mathcal{N}(z_c; \mu_c, \Sigma_c) + \log|\det D f_\theta(y_c; x)|),
\end{aligned} \tag{6}$$

where $C$ denotes the number of camera configurations available in the SIDD dataset.

Through the trained conditional NF, we can synthesize noisy images which have a similar distribution to $y_c$ by taking random samples from $\mathcal{N}(\mu_c, \Sigma_c)$ as an input of the $f_\theta^{-1}$ given ground truth clean image $x$ as the condition. Fig. 1(a) illustrates the proposed framework.

## 4.2 NOISE-AWARE IMAGE GENERATION

Unlike our NF training procedure, we propose to generate noisy images without metadata (*e.g.,* ISO, smartphone manufacturer) given target noisy input $y$. In particular, we aim to synthesize

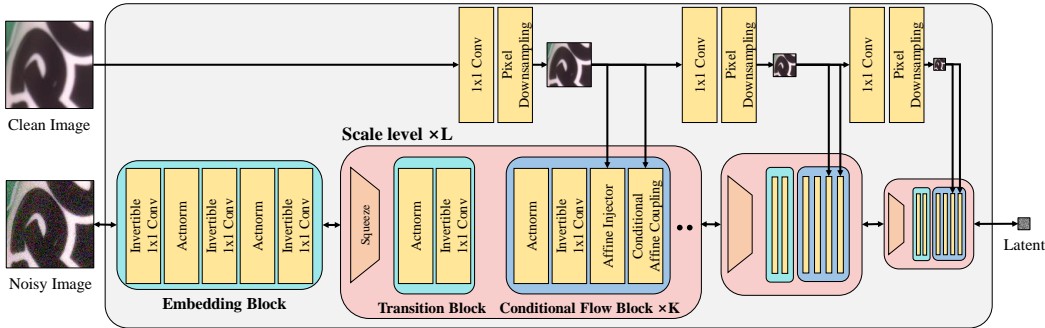

Figure 3: Overview of NAFlow.

noisy images whose noise distributions are similar to the noise distribution of the given input image. To do so, we utilize the capability of the NF as a generative classifier (Mackowiak et al., 2021; Izmailov et al., 2020). Specifically, we obtain a latent $z$ from the noisy input $y$ through the trained NF and measure the likelihood of that latent $z$ for each learned Gaussian distribution (*i.e.,* $\mathcal{N}(z; \mu_1, \Sigma_1), \mathcal{N}(z; \mu_2, \Sigma_2), ..., \mathcal{N}(z; \mu_C, \Sigma_C)$). Recall that we use the GMM in latent space to approximate target noise using a combination of distinct noise information given these likelihoods. Thus, using this set of $C$ likelihood values, we can mix the learned $C$ distinct noise distributions and generate input noise-specific images from the mixture model. The proposed generation process and algorithm are illustrated in Fig. 1(b) and Alg. 1.

---

**Algorithm 1** Noise-Aware Sampling (NAS)

---

**Input:** clean condition $x$, noisy input $y$
1: $z = f_\theta(y; x)$
2: Sample $\tilde{z} \sim \mathcal{N}(\tilde{\mu}, \tilde{\Sigma})$, where $\tilde{\mu} = \sum_c \frac{\mathcal{N}(z; \mu_c, \Sigma_c)}{\sum_{c'} \mathcal{N}(z; \mu_{c'}, \Sigma_{c'})} \cdot \mu_c$, $\tilde{\Sigma} = \sum_c \frac{\mathcal{N}(z; \mu_c, \Sigma_c)}{\sum_{c'} \mathcal{N}(z; \mu_{c'}, \Sigma_{c'})} \cdot \Sigma_c$
3: **return** $\tilde{y} = f_\theta^{-1}(\tilde{z}; x)$

---

In our experiments, we demonstrate that the proposed NAS algorithm 1 allows us to synthesize real-world noisy images without metadata with the aid of our noise-aware generation mechanism.

### 4.3 MULTI-SCALE NOISE EMBEDDING

Taking into account spatial noise correlation in sRGB image is a critical consideration (Zhou et al., 2020; Lee et al., 2022; Fu et al., 2023). Specifically, a majority of digital cameras employ a Bayer filter array, where each pixel in the image sensor can only record the intensity of light for one of the three primary colors (*e.g.,* red, green, or blue), necessitating the interpolation of color information to generate a comprehensive full-color image. In the other words, the process of demosaicking on the Bayer filter induces noise correlation among adjacent pixels (Jin et al., 2020). Moreover, a recent study (Zhang et al., 2023) has computed the size of spatially correlated noise areas within different regions of the image, and found a considerable variance in this area size, ranging from a single pixel to over 25 pixels. Inspired by this noteworthy observation, our NF architecture incorporates a multi-scale noise embedding strategy involving squeeze operations to effectively address different scales of noise correlation. Please note that our approach is the first NF-based attempt to model real-world noise considering noise correlation through the multi-scale embedding approach.

We illustrate our proposed NAFlow in Fig. 3. The overall architecture of our proposed NAFlow draws inspiration from several prior works (Dinh et al., 2017; Kingma & Dhariwal, 2018; Ardizzone et al., 2019; Lugmayr et al., 2020). For a clearer description, we provide brief descriptions of each component of NF in Appx. A.1. The architecture of our model network is designed with three scale-level blocks ($L = 3$) to effectively manage diverse scales of noise. The initial processing step involves encoding the noisy image through an embedding block, which including invertible $1 \times 1$ convolution (Kingma & Dhariwal, 2018) and ActNorm (Kingma & Dhariwal, 2018). Each scale-level block is responsible for accommodating different noise scales and comprises a squeeze operation, a single transition block, and two conditional flow blocks ($K = 2$). It is worth noting that

the squeeze operation reduces the spatial resolution of the input features by half, a crucial step for addressing multi-scale noise. Additionally, we have incorporated an extra transition block following the squeeze operation to mitigate checkerboard artifacts in the reconstructed image, stemming from pixel re-ordering (Lugmayr et al., 2020). We opt for $K = 2$ and bypass conditional flow blocks following the embedding block to transform all channels in the feature at each level, ensuring the network's efficiency. This choice is motivated by the fact that affine coupling can transform only half of the feature channels conditioned on the remaining half (see Appx. A.1). Meanwhile, each conditional block comprises a conditional affine coupling (Ardizzone et al., 2019), an affine injector (Lugmayr et al., 2020), and one transition block. Notably, both the affine injector and conditional affine coupling blocks are conditioned on clean images, enabling them to model noise that contingent upon the underlying clean signals.

## 5 Experiments

To evaluate the capabilities of our noise generation model, we conduct two main experiments. First, we assess the quality of the generated noise quantitatively and qualitatively and investigate noise correlation through visualization (Sec 5.2, Sec 5.3, and Sec 5.4). Second, we examine the effectiveness of the generated noise in real-world denoising tasks by training a denoising network using the generated noisy images (Sec 5.5). Finally, we conduct ablation studies to show superiority of our proposed NAFlow, NAS, and multi-scale noise embedding (Sec 5.6).

### 5.1 Experimental Setup

**Implementation Details.** All the networks are optimized using Adam optimizer (Kingma & Ba, 2014). For training NAFlow, we minimize the $\mathcal{L}_{NLL}$ loss in Eq. 6 with initial learning rate 1e-4 which is reduced by half at 50k, 75k, 90k during 100k iterations. We use randomly cropped patches ($160 \times 160$) and the mini-batch size of 8 for training. For a denoising network, we use the DnCNN (Zhang et al., 2017). Training is conducted with the identical experimental configuration as described in Fu et al. (2023); with 300 epochs, a learning rate $10^{-3}$, and a batch size of 8.

**Dataset.** To train NAFlow, we use SIDD (Abdelhamed et al., 2018) dataset, which has 34 different camera configurations (*i.e., $C = 34$*). Specifically, we use SIDD-Medium split which comprises 320 noisy-clean image pairs captured with five different smartphone cameras: Google Pixel (GP), iPhone 7 (IP), Samsung Galaxy S6 Edge (S6), Motorola Nexus 6 (N6), and LG G4 (G4). To evaluate the noise generation, we use SIDD-validation. The training of the denoiser in Sec. 5.5 also uses the same dataset, but instead of real noisy data, it uses noisy data generated by each noise modeling method. Furthermore, to measure the real denoising performance, we use SIDD-Benchmark dataset.

**Metrics.** To evaluate the quality of the generated noise, we use two metrics: Kullback-Leibler Divergence (KLD) and Average Kullback-Leibler Divergence (AKLD) (Yue et al., 2020). In addition, we adapt PSNR and SSIM metrics to evaluate the denoising performance.

### 5.2 Noise Generation with SIDD

**Compared Baselines.** We compare our model with several noise generation models, including C2N (Jang et al., 2021), Flow-sRGB (Kousha et al., 2022), and NeCA-S, NeCA-W (Fu et al., 2023). We measure noise quality for each camera in the SIDD-Validation to assess device-specific characteristics as in (Fu et al., 2023). Following the usual settings as in (Kousha et al., 2022; Fu et al., 2023), to ensure consistency between the training and validation sets, we ensure that both sets contain the same ISO levels. We use official weight parameters to evaluate the compared methods. Please note that C2N differs from other models in that it trains noise distribution in an unpaired manner, utilizing additional clean dataset and noisy dataset such as BSD500 and DND (Plotz & Roth, 2017) during training.

**Results.** Tab. 2 shows the noise quality in terms of KLD and AKLD conducted on SIDD-Validation of five different cameras. Compared to other noise modeling methods, our NAFlow consistently demonstrates superior performance in terms of average KLD and average AKLD. An important point to note is that our NAFlow shows a performance gain of 0.0035/0.013 in terms of KLD/AKLD compared to NeCA-W (Fu et al., 2023).

Table 2: Quantitative results of synthetic noise on SIDD-Validation. The results are computed with KLD↓ and AKLD↓. In the case of NeCA-S and NeCA-W (Fu et al., 2023), they require five different models for noise generation, one for each smartphone camera. In particular, NeCA-W requires three sub-models (GENet[a], NPNet[b], NCNet[c]) for each camera. Best results are highlighted in **bold**.

| Camera | Metrics | C2N | Flow-sRGB | NeCA-S | NeCA-W | **NAFlow** |
|--------|---------|-----|-----------|--------|--------|-----------|
| G4 | KLD | 0.1660 | 0.0507 | 0.4025 | **0.0242** | 0.0254 |
| | AKLD | 0.2007 | 0.1504 | 1.6803 | 0.1524 | **0.1367** |
| GP | KLD | 0.1315 | 0.0781 | 0.1713 | 0.0432 | **0.0352** |
| | AKLD | 0.1968 | 0.1797 | 0.7379 | 0.1273 | **0.1180** |
| IP | KLD | 0.0581 | 0.5128 | 0.3424 | 0.0410 | **0.0339** |
| | AKLD | 0.2929 | 1.7490 | 1.2924 | **0.1145** | 0.1522 |
| N6 | KLD | 0.3524 | 0.2026 | 0.2830 | **0.0206** | 0.0309 |
| | AKLD | 0.2919 | 0.2469 | 1.0598 | 0.1304 | **0.1108** |
| S6 | KLD | 0.4517 | 0.3735 | 0.1724 | 0.0302 | **0.0272** |
| | AKLD | 0.4190 | 0.2641 | 0.4646 | 0.1933 | **0.1355** |
| Average | KLD | 0.2129 | 0.2435 | 0.2743 | 0.0342 | **0.0305** |
| | AKLD | 0.2802 | 0.5180 | 1.0470 | 0.1436 | **0.1306** |
| #Params | | 2.2M | 6K | $5 * (7.8M^c)$ | $5 * (263K^a + 42K^b + 7.8M^c)$ | 1.1M |

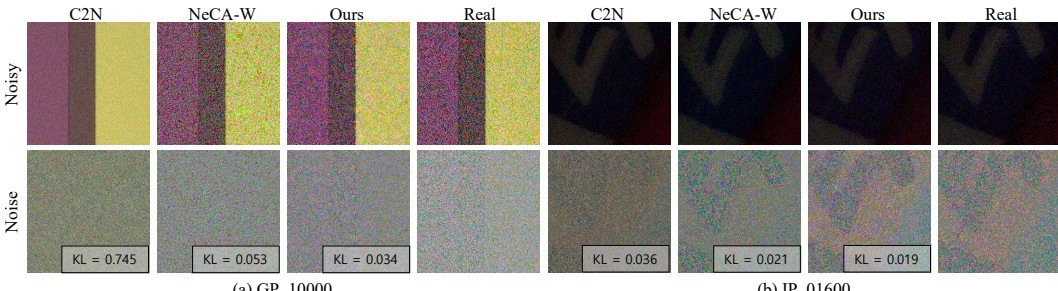

(a) GP_10000          (b) IP_01600

Figure 4: Visualization of synthetic noisy images with different Camera configurations (Camera_ISO) on SIDD-Validation. From left to right: C2N, NeCA-W, Ours (NAFlow), and Real noisy images.

Note that NeCA-W is trained separately for each camera model, so five differently trained NeCA-W models may produce more realistic noise for certain camera models. In contrast, our model uses a single unified model with a much smaller number of parameters for all camera models. Nevertheless, our model shows comparable performance to NeCA-W. Moreover, Fig. 4 compares the generated noise and noisy images. A notable observation is the strong similarity between the noise generated by NAFlow and real-world noise. This indicates that NAFlow excels at synthesizing realistic noise.

## 5.3 NOISE GENERATION WITH SIDD+

Generating noise for different camera configurations that are not included in the SIDD dataset is a challenging task, so most conventional noise generation models synthesize noise using the metadata available in the SIDD. During training, our NAFlow also uses metadata from SIDD (Abdelhamed et al., 2018) to learn a multi-distribution, but unlike other models (Kousha et al., 2022; Fu et al., 2023), NAFlow can generate noise using only noisy-clean image pairs without relying on the metadata during the inference phase. Consequently, NAFlow can generate noise that is similar within the training distribution even for the camera configuration not included in the SIDD dataset. To validate this advantage, we use the SIDD+ (Abdelhamed et al., 2020) dataset, which is captured with smartphones similar to SIDD but with smartphones not used to capture images in the SIDD dataset. Tab. 3 shows the accuracy of synthetic noise using the clean-noise image pairs on the SIDD+ (Abdelhamed et al., 2020). For the pre-trained NeCA-S and NeCA-W models, the challenge is to generate noise for camera configurations which are not available in the SIDD training dataset. Therefore, we show the best KLD values among their five different camera models. In Tab. 3, we can see that NAFlow has the highest performance on SIDD+. This shows that NAFlow is able to generate comparable synthetic noisy images even in the absence of metadata.

Table 3: Quantitative results of synthetic noise from the dataset (SIDD+) not used for training the generative model. The results are computed in terms of KLD↓. Best result is highlighted in **bold**.



(a) C2N   (b) NeCA-W   (c) Ours   (d) Real

| Metric | C2N | NeCA-S | NeCA-W | NAFlow |
|--------|-----|--------|--------|--------|
| KLD↓ | 0.3973 | 0.2620 | 0.0629 | **0.0494** |

Figure 5: Noise correlation map on N6_03200 from different methods. Ours shows the highest similarity with the correlation map in Real (d).

Table 4: Quantitative results of denoising performance on SIDD Benchmark in terms of PSNR↑ and SSIM↑. All methods are trained under the same dataset conditions (synthetic noisy-clean pairs). Note that Real Noise are trained with real noisy-clean pairs. The best result is highlighted in **bold**.

| Metrics | C2N | NoiseFlow | Flow-sRGB | NeCA-S | NeCA-W | **NAFlow** | Real Noise |
|---------|-----|-----------|-----------|--------|--------|------------|------------|
| PSNR↑ | 33.76 | 33.81 | 34.74 | 36.10 | 36.82 | **37.22** | 37.63 |
| SSIM↑ | 0.901 | 0.894 | 0.912 | 0.927 | 0.932 | **0.935** | 0.935 |

## 5.4 Noise correlation visualization

It is important to model real-world noise by considering noise correlation (Fu et al., 2023). In Fig. 5, we compute correlation between the noise in the center pixel and the neighbour noises following the procedure in (Lee et al., 2022; Wang et al., 2023) and compare the noise correlation of our NAFlow with other noise generating networks. Our network generates spatially correlated noise that is most closely resembles the noise correlation in the Real compared to C2N and NeCA-W.

## 5.5 Application: sRGB Denoising

To evaluate the efficacy of the noise generating models and compare their performance in real-world denoising, we train conventional denoising network using synthetic noisy images from C2N (Jang et al., 2021), NoiseFlow (Abdelhamed et al., 2019), Flow-sRGB (Kousha et al., 2022), NeCA-S (Fu et al., 2023), NeCA-W (Fu et al., 2023), and our NAFlow. To be specific, DnCNN is trained with pairs of clean and synthetic noisy images, and denoising performance of DnCNN (Zhang et al., 2017) is compared in Tab. 4. The last column (Real Noise) in Tab. 4 shows the denoising result by DnCNN trained with clean and real noisy image pairs, indicating the upper bound result.

As a result, when using synthetic noisy generated with our NAFlow, DnCNN shows significantly better denoising performance compared to other noise generating methods. Specifically, NAFlow outperforms the current state-of-the-art method NeCA-W by 0.39 dB in PSNR. Note that our NAFlow does not require metadata or a separately trained network for each camera configuration. Comparing the performance with the upper bound, we acknowledge that the PSNR value still lags behind. However, it is worth noting that we significantly reduce the PSNR gap and achieve equivalent performance in terms of SSIM. We also provide denoising results from the differently trained DnCNN in Fig. 6. Our synthetic noise dataset leads to visually more pleasing denoising results, showing higher robustness against remaining noise artifacts compared to the other baseline models.

## 5.6 Ablation Study

**Effect of noise-aware sampling.** As in Alg. 1, our model explores the learned latent space with multiple different noise distributions, and generates input noise-specific noisy images. To evaluate the effectiveness of our noise-aware sampling algorithm, we provide an ablation study result by comparing different sampling methods in Tab. 5. First, the noisy input images are mapped to latents through our NF model. Then we measure the likelihood value of the latents for each normal distribution $\mathcal{N}(\mu_c, \Sigma_c)$, and generate new noisy images using the normal distributions with high likelihood values. In Tab. 5, (Rank-k) indicates the accuracy of the synthetic noise from the normal distribution that has the k-th best likelihood value. As we expect, image samples generated from normal distribution with higher likelihood value give better results, which shows the performance of our NF as a generative classifier.

Table 5: Ablation study on the effective of NAS. We rank the camera configurations $c$ classified by NAFlow and generate the noise from these $c$ and measure the metrics in the SIDD-Validation.

| Metrics | Rank-1 | Rank-2 | Rank-3 | Rank-4 | Rank-5 | Random | **All** |
|---------|--------|--------|--------|--------|--------|--------|---------|
| KLD↓ | 0.0294 | 0.0312 | 0.0408 | 0.0422 | 0.0613 | 0.4700 | **0.0291** |
| AKLD↓ | 0.1302 | 0.1308 | 0.1375 | 0.1463 | 0.1676 | 0.4486 | **0.1293** |

Table 6: Two conditional NFs ($f_\theta$) trained under single latent distribution and multiple latent distributions are compared.

| Latent distribution | KLD↓ | AKLD↓ |
|---------------------|------|-------|
| Single normal distribution | 0.0544 | 0.1657 |
| Multiple distinct distributions | **0.0291** | **0.1293** |

Figure 6: Denoising results in terms of PSNR↑ and SSIM↑ on SIDD-Validation from DnCNN trained on each method. For more qualitative results, please refer to the appendix.

Furthermore, we see that our NAFlow outperforms naive Rank-k sampling, and Random sampling which generates new noisy images from randomly selected normal distribution in the latent.

Table 7: KLD score conditioning on learnable parameters of Gaussian distribution in Eq. 4.

| Training $\mu$ | Training $\Sigma$ | KLD↓ |
|:---:|:---:|:---:|
| ✗ | ✗ | 0.0544 |
| ✓ | ✗ | 0.0449 |
| ✗ | ✓ | 0.0340 |
| ✓ | ✓ | 0.0291 |

Table 8: KLD score depending on different number of scale-level blocks $L$ in Fig. 3.

| Scale-level | KLD↓ |
|:---:|:---:|
| $L = 1$ | 0.0349 |
| $L = 2$ | 0.0313 |
| $L = 3$ | 0.0291 |
| $L = 4$ | N/A |

**Effect of learning multiple distributions.** Our NAFlow is trained by assuming $C$ distinct latent distributions, and the latent sampling mechanism presented in Alg. 1 allows us to generate input noise-aware noisy images without additional metadata. To show the superiority of our generation mechanism considering multiple distinct distributions, we train the NF ($f_\theta$) by assuming a single distribution (*i.e.*, $\mu_c = 0$, $\Sigma = \mathbf{I}$ for all $c$) and compare the accuracy of the generated noisy images in terms of KLD. In Tab. 6, we see that NAS based on multiple different distributions renders more accurate noisy images on the SIDD-Validation.

**Trainable parameters of Gaussian distribution.** NAFlow leverages multiple Gaussian models with both trainable mean and covariance in the latent space, effectively addressing the variety of real-world noise modeling as described in Eq. 4. In Tab. 7, we conduct an ablation study to assess the effectiveness of each parameter of the Gaussian distribution within the latent space. Our findings indicate that the NAFlow with learnable both mean and covariance of Gaussian distribution outperforms the other models. This indicates that the richer expressive power of the Gaussian distribution is crucial in accurately modeling the various real-world noise.

**Impact of multi-scale noise embedding.** In our flow architecture, we integrate a multi-scale noise embedding approach to handle spatial noise correlations of varying areas. As shown in Tab. 8, an increase in the scale-level leads to better KLD scores on the SIDD-Validation, indicating our model's ability to adeptly learn multi-scale noise characteristics. However, we could not investigate the higher scale level factor (e.g., $L \geq 4$) due to unstable training.

## 6 CONCLUSION

In this study, we developed a novel NAFlow framework to address the complex and diverse noise distributions that arise from a variety of camera settings. In addition, we introduced the Noise-Aware Sampling (NAS) algorithm, which builds on the foundation of the NAFlow framework. NAS allows our noise generation model to use Gaussian mixture models derived from multiple noise distributions to synthesize realistic sRGB images with noise. A notable advantage of our method is that it eliminates the need to input noisy image metadata during inference. In addition, we recognize the importance of considering noise correlations with different scales. Consequently, we present a multi-scale noise modeling approach that effectively addresses noise correlation across a range of scales. The experimental results demonstrate the significant performance improvement that our algorithm brings to both the noise modeling and the denoising network, highlighting the effectiveness of our proposed methods.

ACKNOWLEDGMENTS

This work was supported by Institute of Information & communications Technology Planning & Evaluation (IITP) grant funded by the Korea government(MSIT) (No.2022-0-00156, Fundamental research on continual meta-learning for quality enhancement of casual videos and their 3D meta-verse transformation) and Institute of Information & communications Technology Planning & Evaluation (IITP) grant funded by the Korea government(MSIT) (No. RS2023-00220628, Artificial intelligence for prediction of structure-based protein interaction reflecting physicochemical principles, 30%)

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

# A APPENDIX

## A.1 NAFLOW ARCHITECTURE DETAIL

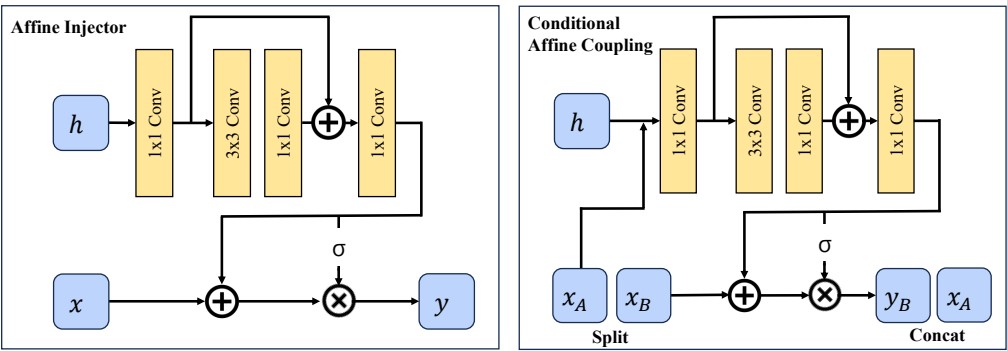

Figure 7: Forward pass of Conditional Affine Coupling and Affine Injector.

The overall architecture of our proposed NAFlow draws inspiration from several prior works (Dinh et al., 2017; Kingma & Dhariwal, 2018; Ardizzone et al., 2019; Lugmayr et al., 2020).

**Conditional Affine Coupling** (Ardizzone et al., 2019): The conditional affine coupling layer (Ardizzone et al., 2019) is an extension of the concept of an affine coupling layer as originally proposed in the RealNVP (Dinh et al., 2017). The distinguishing feature of a conditional affine coupling layer is that it depends on a conditional input to perform an affine transformation. Given an input $x$ and a conditional input $h$, an affine coupling layer splits it into two parts: $x_1$, $x_2$. Note that the conditional input $h$ is a downsampled clean image by the pixel downsampling operator in our NAFlow. One part $x_1$, remains unchanged, while the other part $x_2$, undergoes an affine transformation. The transformation can be defined as follows:

$$\begin{aligned} y_1 &= x_1 \\ y_2 &= x_2 \odot \exp(f_s(x_1; h)) + f_t(x_1; h), \end{aligned} \tag{7}$$

where $f_s$ and $f_t$ are arbitrary neural networks to extract scale and translation factors to perform affine transformations, and $\odot$ indicates element-wise multiplication. In Fig. 7, we compose a few layers of $1{\times}1$ convolution and $3{\times}3$ convolution to design the neural networks $f$ to maintain the simplicity and efficiency of our NAFlow. And the inverse transformation can be defined as follows:

$$\begin{aligned} x_1 &= y_1 \\ x_2 &= (y_2 - f_t(x_1; h)) \oslash \exp(f_s(x_1; h)), \end{aligned} \tag{8}$$

where $\oslash$ stands for element-wise division. Finally, the log determinant of this transformation can be calculated as $\log|\det D f_s(x_1; h))|$.

**Affine Injector** (Lugmayr et al., 2020): To further increase the influence of the conditional input on the mapping performed by NF, an affine injector is introduced to transform input features solely based on conditional input features. We adopt this layer to inject clean signal information on each scale block in Fig. 3 because learning signal dependence is the important factor in modelling real-world noise. The difference between Affine Injector and Conditional Affine Coupling is that it uses the output features of neural networks, which only use the conditional input $h$, as follows:

$$\begin{aligned} y_1 &= x_1 \\ y_2 &= x_2 \odot \exp(f_s(h)) + f_t(h). \end{aligned} \tag{9}$$

And for the inverse process of the layer, each element-wise multiplication and addition operation is replaced by an element-wise division and subtraction operation.

**Invertible 1$\times$1 Convolution** (Kingma & Dhariwal, 2018): The operation is introduced to generalize the channel-wise permutation (Dinh et al., 2017; 2015) and is fully invertible since it independently multiplies each spatial coordinate of the input feature. We employ LU decomposition (Kingma & Dhariwal, 2018) to factorize the weight matrix of the $1{\times}1$ Convolution to reduce the computation of its determinant and enhance model training stability (Wolf et al., 2021).

**Actnorm** (Kingma & Dhariwal, 2018): The operation normalizes the input feature along the channel axis using learned affine transformation parameters.

**Squeeze**: Having the same structure as the pixel downsampling operator (Shi et al., 2016), the squeeze operation is invertible and reduces the resolution of the input feature in half by moving each spatial $2 \times 2$ neighborhood of pixels into the channel axis and allowing for learning broader spatial correlations. To follow the naming convention in previous NF works, we refer to the operation as Squeeze if it is used in NF module.

## A.2 QUALITATIVE RESULTS OF NOISE GENERATION

In Fig. 8, we present additional visualization results of noise sampling using our method (NAFlow), as well as comparisons to existing methods (Jang et al., 2021; Fu et al., 2023). Based on the noise (bottom of each row in Fig. 8), we can confirm that NAFlow is the most similar to the real noise. In the case of C2N (Jang et al., 2021), it shows similar results in terms of KLD for low-light images, but it does not consider noise correlation. NeCA (Fu et al., 2023) demonstrates superior results compared to C2N, however it is clear that ours (NAFlow) outperforms existing methods in terms of noise correlation and KL metric.

## A.3 QUALITATIVE RESULTS OF DENOISING PERFORMANCE

We present additional denoising results using DnCNN (Zhang et al., 2017) denoiser. We also provide denoising results from C2N (Jang et al., 2021) and NeCA (Fu et al., 2023) for comparison. The DnCNN denoisers and related test code in C2N and NeCA use pretrained official weight parameters and code from the official GitHub repositories.

- C2N (Jang et al., 2021): https://github.com/onwn/C2N
- NeCA (Fu et al., 2023): https://github.com/xuan611/sRGB-Real-Noise-Synthesizing

Based on the visualization in Fig. 9, it is apparent that NAFlow (Ours) surpasses other models in regards to PSNR and exhibits the highest similarity to the clean image when compared.

## A.4 NOISY GENERATION FROM METADATA

We aim to generate noise without metadata, but it is possible to generate noise using only clean images learned from known distributions for each camera configuration. In Fig. 10, images generated from clean images using three cameras (GP, N6, S6) and five different ISO (100, 400, 800, 1600, 3200) settings are provided. We can observe that even at the same ISO level, slightly different noise is generated depending on the camera (sensor) model.

## A.5 LIMITATION OF THE SIDD DATASET

The limited range of smartphone model types and lack of scene diversity in the SIDD (Abdelhamed et al., 2018) dataset makes it challenging to generalize to real-world noise modeling. Consequently, real-world noise modeling studies face difficulties in utilizing SIDD for their methods, and there is a need for more large-scale real-world noise datasets for better understanding of real-world noise.

Furthermore, the sRGB images in the SIDD dataset are acquired using the simplified RAW-to-sRGB pipeline[1] and it can produce the distribution gap from noise in real-world images. Therefore, these limitations should be addressed in future research.

---

[1]https://github.com/AbdoKamel/simple-camera-pipeline

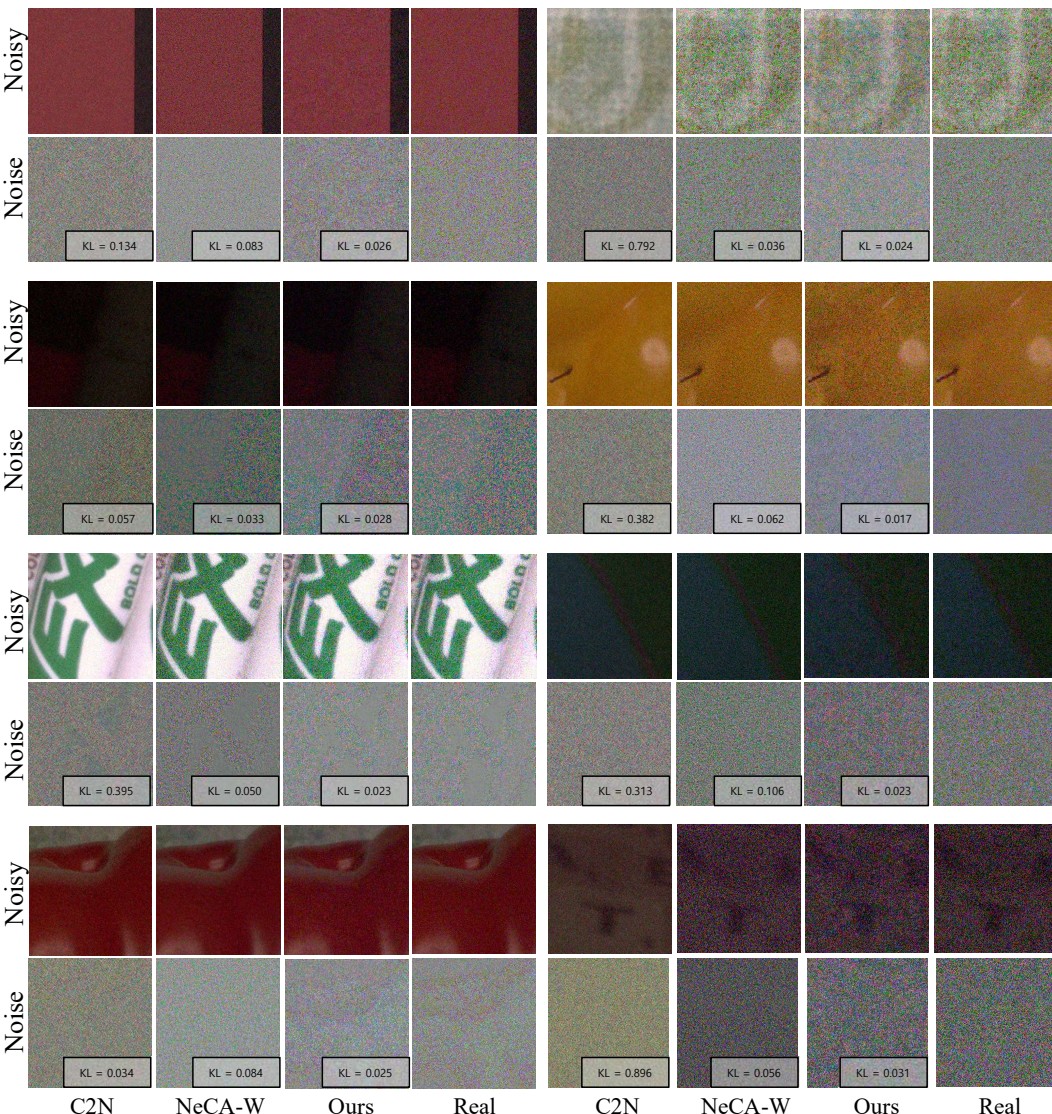

Figure 8: Visualization of synthetic noise images on SIDD-Validation. The images are arranged from left to right in the following order: C2N, NeCA-W, Ours (NAFlow), and Real noisy image. For more detail, noise images are also provided at the bottom of each row.

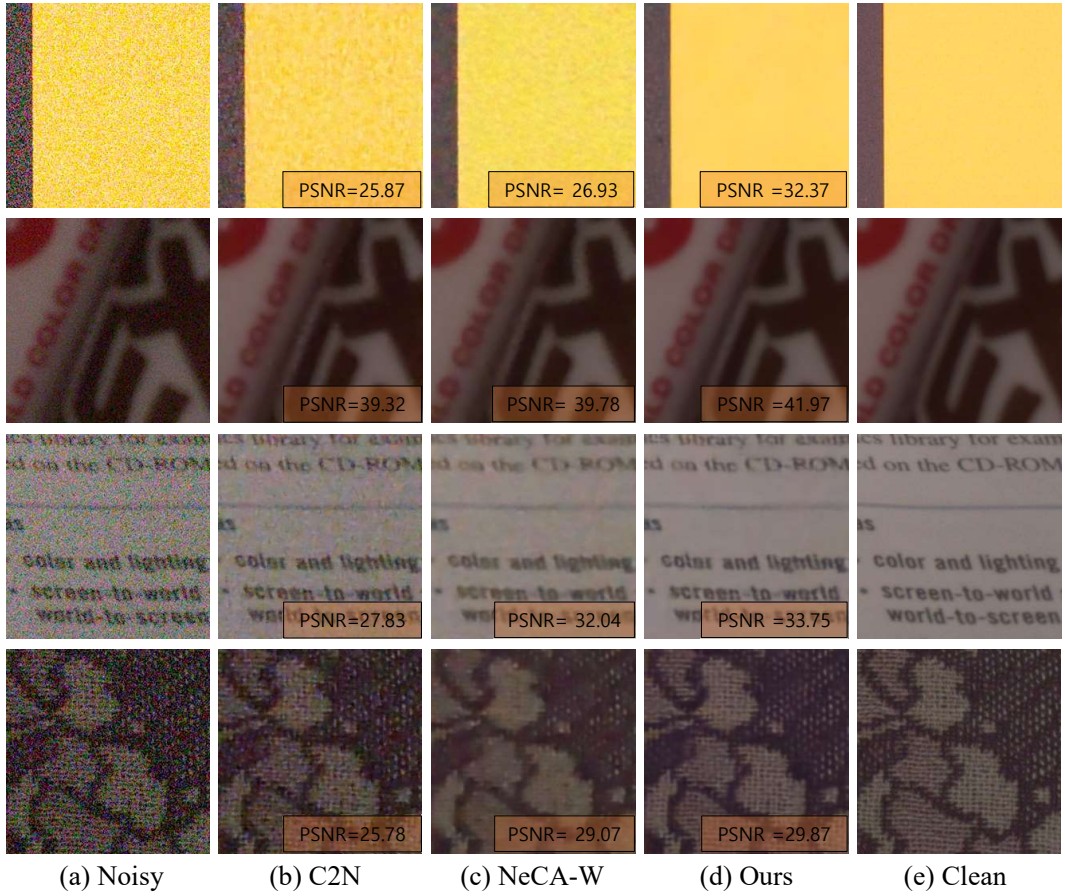

(a) Noisy     (b) C2N     (c) NeCA-W     (d) Ours     (e) Clean

Figure 9: Denoising results on SIDD-Validation from DnCNN trained on each method.

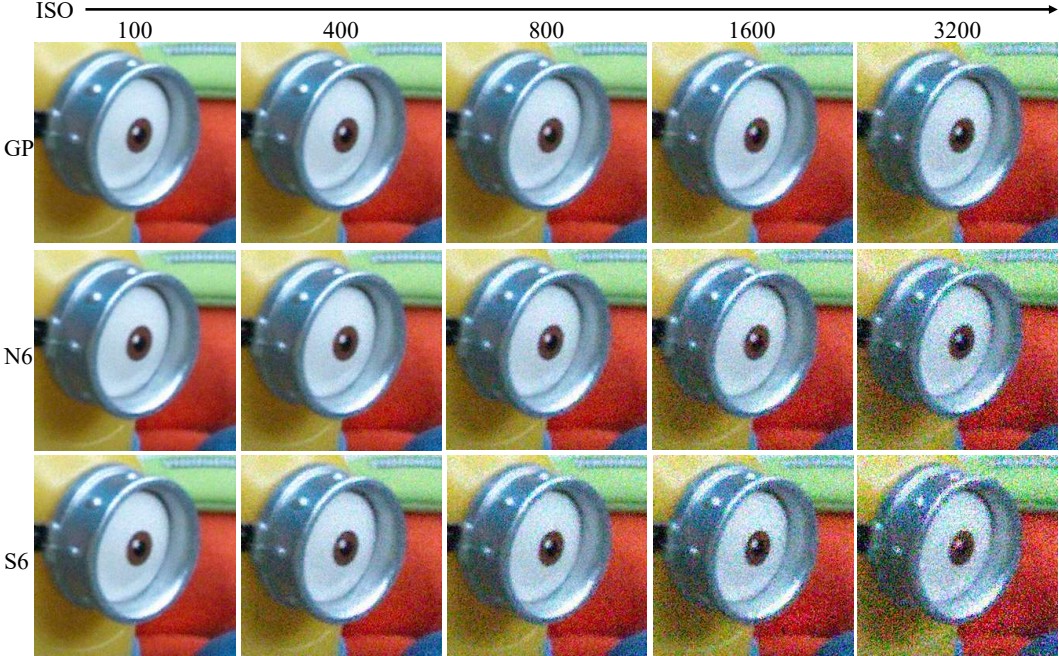

Figure 10: Generation results using camera configuration of NAFlow. The images is obtained from SIDD+ (Abdelhamed et al., 2020).

