# OpenReview forum: "sRGB Real Noise Modeling via Noise-Aware Sampling with Normalizing Flows"
_ICLR.cc/2024/Conference — ICLR 2024 poster_

### Official Review · Reviewer_VvGJ · 2023-10-18

**Soundness:** 3 good
**Presentation:** 3 good
**Contribution:** 3 good
**Rating:** 6
**Confidence:** 5

**Summary:**

The paper introduces a new normalizing flow framework (i.e., NAFlow) for noise modeling and synthesis. For noise modeling, NAFlow learns to map noise of various smartphone types and gain settings to different Gaussian distributions. For noise synthesis, NAFlow uses the Gaussain mixture model of learnt distributions to generate accurate yet diverse noisy images. Experiment results on real-world denoising datasets show the superiority of NAFlow over existing methods.

**Strengths:**

1. The idea of learning the noise distribution implicitly without metadata is reasonable. It is practical to model the noise of devices that is hard to accquire metadata, such as smartphones.
2. The proposed NAFlow outperforms baseline methods by a large margin on both noise modeling and noise synthesis.
3. The paper is well organized and written.

**Weaknesses:**

1. The comparison with C2N is unfair. C2N is trained with only noisy images, while other methods are trained with paired images. This training setting should be pointed out in the experiments.

**Questions:**

1. (This question is not critical to rating) What are the applications of noise synthesis and modeling methods? These methods apply paired noisy-clean images for training, then synthesize new pairs from clean images for training denoiser. However, the denoiser trained with synthesized pairs shows inferior performance than trained with original real paires. Could the authors explain the applications of proposed method?

---

> ### Author Response · Authors · 2023-11-16
>
> Dear Reviewer VvGJ,
>
> We would like to thank the reviewer for taking time to review our manuscript and acknowledging our contributions. Below is our response to reviewer’s concerns.
>
> ---
>
> > Comparison with C2N
>
> We acknowledge that comparing C2N with other sRGB noise modeling methods is unfair. **C2N differs from other models in that it learns noise and clean data in an unpaired manner, utilizing additional clean dataset (e.g., BSD500) and noisy dataset (e.g., DND) during training.** However, due to the limited availability of comparable sRGB noise modeling methods, we have included C2N for comparison, as demonstrated by the previous method (NeCA). Nevertheless, as the reviewer pointed out, we have specified the training dataset of C2N in the revision (Sec 5.2).
>
> > Applications of noise synthesis and modeling method
>
> The main purpose of noise modeling is to understand noise properties and to denoise a distorted image for aesthetic purposes or improving downstream tasks (e.g., object detection, segmentation). For RAW images, the prior works [5\*, 6\*] outperform the denoising results using only the synthesized noisy-clean pairs over the real noisy-clean pairs. **However, sRGB noise is further distorted due to several non-linear operations during the RAW-sRGB conversion, making it more difficult for sRGB noise modeling to outperform denoising performance using real data.** To alleviate the problem, prior studies blend synthetic noise datasets with the real noise datasets to enhance the denoiser’s performance [7\*] or generating synthetic noise datasets using additional clean datasets [8\*]. We believe that more improved results can be achieved through a better understanding of sRGB noise and improved technical contributions in future works.
>
> > Reference
>
> [5\*] Abdelhamed, Abdelrahman, et al. "Noise flow: Noise modeling with conditional normalizing flows." ICCV. 2019.
>
> [6\*] Maleky, Ali, et al. "Noise2noiseflow: realistic camera noise modeling without clean images." CVPR. 2022.
>
> [7\*] https://github.com/GarrickZ2/Image-Denoising
>
> [8\*] Yuanhao Cai, et al. “Learning to generate realistic noisy images via pixel-level noise-aware adversarial training.” NeurIPS. 2021.

---

> > ### Comment · Reviewer_VvGJ · 2023-11-18
> >
> > The authors have addressed my concerns, thanks for detailed response.

---

> ### Author Response · Authors · 2023-11-18
>
> We appreciate your valuable feedback and are pleased to hear that your concerns have been addressed. Thank you for your thoughtful review and support.

---

### Official Review · Reviewer_s3bG · 2023-10-27

**Soundness:** 3 good
**Presentation:** 3 good
**Contribution:** 3 good
**Rating:** 8
**Confidence:** 5

**Summary:**

This work proposes a normalizing flow-based framework NAFlow for realistic noisy image generation. The generated images are used to adapt real denoisers and boost their performance. With the proposed Noise-Aware sampling algorithm, NAFFow learns and effectively manages the diverse noise distributions originating from various camera settings.

**Strengths:**

(i) The idea of using the Normalizing Flow to model the in-camera setting distribution is interesting. Previous methods mainly focus on designing a generative adversarial network for noise modeling.

(ii) The results on self-supervised image denoising are good and solid.

(iii) The presentation is good and easy to follow.

(vi) The appendix provides lots of visual results to show the advantages of the proposed method.

(v) The ablation study is sufficient to demonstrate the effectiveness of the approaches.

**Weaknesses:**

(i) The motivation is unclear. Why using Normalizing Flow is not explained. Because using GAN, Auto-encoder, DeFusion Model, etc., can also model the noise distribution. So, what's the difference?

(ii) The paper proposes a normalizing flow (NF) for noise generation. However, normalizing flow for noise generation has been proposed in NoiseFlow. What is the difference and your contributions?

(iii) The proposed method has some technical drawbacks.

[1]  The normalizing flow-based framework requires a statistic, i.e., the mean of the specific normal distribution for the configuration c. This is a very strong priori condition. In real application, this statistic may not be given.

[2] The proposed method is trained on SIDD, even the pre-trained denoiser. However, the noise distribution varies across different datasets (like DND, PloyU, Nam) since they are captured by different hardware. The training may make the noise generator overfit on the SIDD dataset and not fit well with other noise distributions. Then how to solve this issue?


(iv) The experiments need to be improved.

[1] Experiments on other noisy image datasets should be provided to resolve my concerns in (iii) [2].

[2] The comparisons should also be improved. The metric KLD may not be suitable for noise generation. Why not follow the metric PSNR Gap in DANet or Maximum Mean Discrepancy (MMD) to measure the domain gap?

[3] The adaptation now is conducted in a self-supervised manner. What about its performance in a fully supervised manner? And what about its performance on the SOTA denoising methods like Restormer? Since some noise generation methods like PNGAN can still improve the performance of Restormer.

[4] The comparison between the proposed NAFlow and other SOTA noise generation algorithms should be added in the main paper, including DANet, PNGAN, GDANet, CycleISP, C2N, etc.

(v) Reproducibility: The network and the method (Meta-train and Fast Adaptation) are pretty complicated. It is difficult for other researchers to reproduce the whole method. Since the code and models are not provided, the reproducibility cannot be checked.

**Questions:**

What are the advantages of your method when compared with other NF image denoising algorithms? Show experiments to support this.

---

> ### Author Response · Authors · 2023-11-16
>
> Dear Reviewer s3bG,
>
> We would like to thank the reviewer for the thorough examination on our work. Below is our response to the reviewer's concerns.
>
> ---
>
> > Motivation of choosing Normalizing Flow
>
> We have chosen the normalizing flow (NF) architecture based on three primary considerations: **1. precise likelihood estimation capabilities**, **2. allowance for different noise distributions in latent space** and **3. efficiency**. Our proposed NAFlow demonstrates the capacity to manage multiple noise distributions using a single unified network and can synthesize noise without relying on noise metadata. This distinguishes it from other generative models, such as autoencoders or GANs, which face challenges in producing input-noise-aware results in the absence of metadata. And we also note the efficiency of our approach in comparison to diffusion-based models, which incur higher computational costs due to multiple denoising steps for generation. Furthermore, we would also like to point out that our NAFlow has x40 fewer model parameters compared to NeCA, the current GAN-based SOTA model, while maintaining competitive performance (Tab 2, 4).
>
> > Statistic of Normal distribution in latent space
>
> In our framework, **the mean and variance of normal distributions are also trainable parameters** (Sec 4.1), and we can measure the likelihoods between the latent distribution of the input data and the learned normal distribution with the learned mean and variance parameters at test time (Sec 4.2). In addition, we perform ablation studies to evaluate the effectiveness of each parameter and demonstrate the expressiveness of Gaussian Mixture Models (GMMs) in modeling diverse noise distributions within the latent space (Tab 7). Specifically, we observe that the noise embedded in the normal distribution with trainable mean and variance achieves better noise quality, and it implies that the normal distribution with flexible statistic helps our network to handle diverse real-world noise distributions.
>
> > Response to (iii)-[2]
>
> `The proposed method is trained on SIDD, even the pre-trained denoiser.`
>
> **We do not use pre-trained denoisers for denoising experiments.** All denoisers are trained from scratch (Sec 5.5).
>
> > Overfitting problem on SIDD dataset
>
> **We acknowledge the limitations of the SIDD dataset (Sec A.5),** which contains a limited number of captured scenes and smartphone device types, and it can lead to an overfitting problem. Consequently, achieving generalized performance is a challenging problem for all models trained on the SIDD dataset. **To demonstrate a better generalization capability of NAFlow, we conducted additional experiments on the SIDD+ dataset, which is not included in the training dataset** **in Table 3** of the main manuscript. Note that the SIDD+ dataset consists of new smartphone device types (e.g., HTC U12+, LG G7) and different scenes that were not included in the original SIDD dataset. In Table 3, NAFlow shows better performance compared to the other SOTA models such as NeCA on the external dataset, owing to the proposed noise-aware sampling, which allows NAFlow to approximate the input noise distribution fully facilitating learned noise information without requiring any metadata. See the table below for more results.
>
> > Evaluation on different datasets
>
> Following the reviewer's feedback, **we evaluated the quality of the generated noise on additional datasets (e.g., DND, PolyU, Nam) in terms of KLD**. For the DND dataset, it should be noted that due to the lack of clean images, measuring KLD is not feasible, so we provide results on the SIDD+, PolyU, and Nam datasets.
>
> In the table below, we provide experimental results from the same setup as described in Sec 5.3 (Noise Generation with SIDD+). As the reviewer pointed out, Nam and PolyU, which are the noisy-clean  paired datasets captured by DLSRs, contain noise distributions different from the SIDD dataset because DLSRs have different hardware specifications (e.g., image sensor type) and image processing pipelines. As a result, the noise quality is degraded on Nam and PolyU dataset due to the domain gap and improving real-world noise modeling on diverse environment is our future research direction.
>
> | Metric | SIDD+ | PolyU | Nam |
> | --- | --- | --- | --- |
> | KLD ↓ | 0.0494 | 0.2304 | 0.2055 |

---

> ### Author Response · Authors · 2023-11-16
>
> > Additional noise quality metrics
>
> While Table 3 in the manuscript illustrates the PSNR differences between denoisers trained with real noise and synthesize noise, as you suggested, we have rephrased PSNR Gap for clarity, and measure the noise quality in terms of MMD. As a result, **NAFlow demonstrates robust performance in terms of MMD.** For PNGAN, official weight parameters were not provided, so we report the results based on the results provided in the paper and the official GitHub repository of PNGAN [3*].
>
> | Method | DANet | PNGAN | NeCA-S | NeCA-W | Ours (NAFlow) |
> | --- | --- | --- | --- | --- | --- |
> | PSNR Gap ↓ | 2.06 | 0.84 | 1.53 | 0.81 | **0.41** |
> | MMD ↓ | 0.020 | 0.033 | 0.034 | 0.021 | **0.017** |
> | AKLD ↓ | 0.212 | 0.153 | 1.047 | 0.144 | **0.131** |
>
> > The reason for not using MMD
>
> MMD was employed as a noise quality metric used in PNGAN. For the evaluation, **PNGAN randomly sampled 100 images from both real noisy and synthetic noisy image sets and measured the difference between the two distributions using MMD.** Instead of using this approach, we opt to directly compare the entire image set following the evaluation metrics employed by NeCA, the state-of-art sRGB noise modeling network. We made this decision to ensure a fair comparison with NeCA and other conventional approaches such as C2N and Flow-sRGB.
>
> > Response to (iv)-[3]
>
> `The adaptation now is conducted in a self-supervised manner.`
>
> **Our method does not include adaptation process and self-supervised training.**
>
> > Fully supervised result in Restormer
>
> We did not include the comparisons against PNGAN due to fairness concerns. Because, in the process of learning and generation noise, PNGAN uses an external dataset and a pre-trained denoiser (RIDNet). Regardless, following the suggestions by reviewer, when our model is trained using the same approach, the results are as follows:
>
> | Method | Trainset | PSNR / SIMM |
> | --- | --- | --- |
> | Restormer | SIDD real clean-noisy pair | 40.0155 / 0.9602 |
> | PNGAN+Restormer | + external real clean-synthetic noisy pair | 40.0205 / 0.9603 |
> | Ours+Restormer | + synthetic noisy pair | **40.0405 / 0.9603** |
>
> PNGAN does not provide pretrained weights on the official GitHub repository, so we have filled in the performance values listed on the official GitHub [3*, 4*] In the above table. We use our NAFlow to generate noisy images and trained Restormer by mixing SIDD real noisy datasets with synthetically generated noisy datasets, similar to PNGAN, and can see slightly better performance than the results from PNGAN.
>
> > Comparison between other SOTA noise generation algorithms
>
> We conducted a focused comparison with state-of-the-art methods in noise modeling methods, NeCA. So, we use the same comparison methods as used in NeCA. **As the reviewer suggested, we also compare the** **performance** **of the SOTA noise generation methods in terms of AKLD on the SIDD dataset.**
>
> | Method | DANet | GDANet | CycleISP | C2N | PNGAN | NeCA-W | Ours |
> | --- | --- | --- | --- | --- | --- | --- | --- |
> | AKLD ↓ | 0.212 | 0.253 | 0.716 | 0.213 | 0.153 | 0.144 | **0.131** |
>
> Additionally, we provide device-specific KLD measurement results in our paper, as shown in Table 2. As mentioned above, PNGAN does not provide official pre-trained weights, and since neither the paper nor the official GitHub provides device-specific KLD values, we left these entries blank.
>
> | Smartphone Model | DANet | GDANet | CycleISP | C2N | PNGAN | NeCA-W | Ours |
> | --- | --- | --- | --- | --- | --- | --- | --- |
> | G4 | 0.1081 | 0.1653 | 0.3006 | 0.1660 | - | **0.0242** | 0.0254 |
> | GP | 0.2015 | 0.3136 | 0.2803 | 0.1315 | - | 0.0432 | **0.0352** |
> | IP | 0.0602 | 0.1700 | 0.3732 | 0.0581 | - | 0.0410 | **0.0339** |
> | N6 | 0.0932 | 0.2513 | 0.3893 | 0.3524 | - | **0.0206** | 0.0309 |
> | S6 | 0.0794 | 0.1467 | 0.4407 | 0.4517 | - | 0.0302 | **0.0272** |
>
> > Response to (v)
>
> `The network and the method (Meta-train and Fast Adaptation) are pretty complicated.`
>
> **Our paper does not include any meta-training and fast adaptation.**
>
> > Reproducibility
>
> The code will be available upon acceptance.

---

> > ### Author Response · Authors · 2023-11-16
> >
> > > Contribution & key advantages of proposed method
> >
> > In the overall context of our main manuscript, **we conduct experiments to demonstrate the superiority of our approach by comparing it with existing noise modeling methods**, including the state-of-the-art NeCA, and Flow-sRGB which is the NF-based SOTA model. To provide clarity, we outline the experiments in the main manuscript:
> >
> > - **Table 1** highlights distinctive advantages of our method, compared with Flow-sRGB and NeCA.
> > - **Table 2** shows the quality of the generated noise using KLD and AKLD for each device type in SIDD, demonstrating the superior performance of NAFlow over both Flow-sRGB and NeCA.
> > - Furthermore, in **Table 3**, we experimentally demonstrate the effectiveness of our noise quality on SIDD+ without requiring metadata, where this is the distinguishing feature from previous approaches.
> > - In **Table 4**, we show significant improvements in performance through training the denoiser with synthetic noisy data. Our results demonstrate superior performance relative to other pre-existing models.
> > - **Tables 5**, **6**, and **7** present ablation studies on our architecture's key features and modules, providing the effectiveness of the proposed methods in NAFlow.
> >
> > Furthermore, we also summarize key differences with Noise Flow and our method in "Response to ALL reviewers" section. Please refer to this section.
> >
> > >Reference
> >
> > [3\*] https://github.com/caiyuanhao1998/PNGAN
> >
> > [4\*] https://github.com/GarrickZ2/Image-Denoising

---

### Official Review · Reviewer_EqxC · 2023-10-30

**Soundness:** 3 good
**Presentation:** 3 good
**Contribution:** 3 good
**Rating:** 6
**Confidence:** 4

**Summary:**

This paper proposes a sRGB real noise modeling based on noise-aware sampling and normalized flow. Specifically, the proposed method includes (1) a new NAFlow framework with the Noise-Awar algorithm (NAS), which allows the use of Gaussian mixture models from multiple noise distributions to synthesize real sRGB images with noise without the need for metadata. (2) a multi-scale modeling method to capture the noise correlations with different scales.

**Strengths:**

1. The paper looks technically sound and describes the algorithm clearly.
2. Experimental results demonstrate the advantages of NAFlow compared to some previous methods.
3. The Noise-Aware Sampling (NAS) algorithm eliminates the need to input noisy image metadata during inference which is novel to me.

**Weaknesses:**

1.This paper combines the normalized flow generation of Flow-sRGB with the multi-scale spatial correlation of NeCA. In my opinion, the author draws more on the flow-normalized noise generation method of Flow-SRGB and further uses multi-scale noise modeling to improve performance. I want to know the effect of adding multi-scale noise modeling on Flow-sRGB, and related ablation experiments should be added.
2.Writing errors, such as “NECA-W”, ” NAflow”, et al.

**Questions:**

See Weaknesses

---

> ### Author Response · Authors · 2023-11-16
>
> Dear Reviewer EqxC,
>
> We would like to thank the reviewer for the thorough examination on our work. Below is our response to the reviewer's concerns.
>
> ---
>
> > Multi-scale noise modeling performance on Flow-sRGB
>
> | Model / Scale-level | L=1 | L=2 | L=3 |
> | --- | --- | --- | --- |
> | Flow-sRGB | 0.2436 | 0.2346 | 0.2205 |
> | NAFlow (ours) | **0.0349** | **0.0313** | **0.0291** |
>
> We observe that modeling real-world noise with a multi-scale embedding, as presented in our main manuscript, also improves the performance of Flow-sRGB. We evaluate the similarity of the basic noise characteristics using the KLD metric on SIDD validation set, the experiments presented in the table above specifically show the impact of multi-scale embedding on Flow-SRGB. However, **despite the fact that NAFlow uses a single unified model without the need for metadata, NAflow consistently produces better performance than Flow-sRGB across all scale levels.**
>
> Although modeling real-world noise with a multi-scale embedding may bring improvements to other models, such as Flow-sRGB, **we argue that the major factor in the performance improvement in our work is caused by learning the multiple different noise distributions in the latent space and noise-aware sampling (Tab 5, 6, 7).** For a more detailed comparison, please refer to the "Response to ALL reviewers" section, which highlights the key differences with NeCA and Flow-sRGB.
>
> > Typos
>
> Thanks to your feedback, we have rectified the typos on the revision (Sec 5, Tab 2)

---

> ### Comment · Reviewer_EqxC · 2023-11-16
> **Reply to Authors**
>
> Thanks for detailed response. The authors have solved my concerns. Hence, the rating has been updated.

---

> ### Author Response · Authors · 2023-11-17
>
> Thank you for your thorough feedback and positive reassessment. We appreciate your time and consideration in reviewing our work.

---

### Official Review · Reviewer_YHPu · 2023-10-30

**Soundness:** 3 good
**Presentation:** 4 excellent
**Contribution:** 3 good
**Rating:** 6
**Confidence:** 4

**Summary:**

This paper proposes a system to generate realistic sRGB noise with noiseflow. The idea is to encode noise from multiple cameras into the same common latent space. For an unseen camera, a sample noise is encoded and a new noise is draw from a gaussian mixture model of learned noise. Multi-scale encoding schemes are employed to model long range spatial correlation of noise. Experiments show that a denoising network trained on generated noise works better than comparison techniques. Ablation study is provided and shows the importance of each of the proposed components.

**Strengths:**

I think the technical contribution of this work is solid. The proposed method works well with multiple camera models without the need for retraining. With different proprietary processing, noises from different cameras can have very different characteristics. The idea to encode different noise into different gaussian in a common latent space is convincing, and the result seems to suggest that this works well. The comparison was done convincingly to account for differences in how the comparison methods should be applied (for example, the best value was reported for the NeCA models where it needed to be trained per-camera).

**Weaknesses:**

Despite being successful in learning noise from SIDD dataset, I find the noise being generated to be far from realistics. Modern cellphone cameras apply heavy denoising in the chrominance channel that chromatic noise is never visible. Examples shown throughout the paper (e.g. in figure 4) shows heavy chromatic noise. Further, SIDD dataset only contain images from 10 scenes, so there is not a lot of diversity in the data. This limits how much generalization we can expect to real world cameras. While this is largely a limitation of the SIDD dataset, I think it is very important for the author to acknowledge this point in their manuscript.

As an alternative, it may be interesting for the authors to look at other dataset. There is a technical report by Jaroensri et al. (“Generating Training Data for Denoising Real RGB Images via Camera Pipeline Simulation”) that provides a dataset of raw-processed pairs that, in my opinion, is the most realistic among the raw-processed datasets available. The author should also consider applying processing from raw-only datasets such as those from Chen et al (“Learning to See in the Dark”) to achieve a more realistic noise distribution. Having included more dataset (especially in evaluation) could also be helpful in providing signal for generalization to the real world.

**Questions:**

Please see my weakness section.

---

> ### Author Response · Authors · 2023-11-16
>
> Dear Reviewer YHPu,
>
> We would like to thank the reviewer for recognizing our contribution and for pointing out the specific strength of our method, as well as constructive concerns. We address the reviewer's concerns as follows.
>
> ---
>
> > Limitation of the SIDD dataset
>
> We also acknowledge the limitations of the SIDD dataset as suggested by the reviewer and have reflected this in the revision (Sec A.5). We appreciate the valuable feedback.
>
> > Evaluation on external datasets
>
> **As the reviewer recommended, we have extended our experiment by training NAFlow on the SID[1\*] dataset.** We select 10 camera configurations (C=10) that have a sufficient number of images for training from the SID dataset, and since the SID dataset does not provide noise-free images, we adapt the denoiser [2\*] pre-trained on the SIDD dataset to produce pseudo-ground truth images. In the table below, we measure the similarity between the real noise and our synthesized noise using KLD and AKLD, demonstrating that our method performs similarly on the Fuji dataset while showing a performance gap on the Sony dataset when compared to the SIDD dataset. The drop in performance can be attributed to the fact that the SID dataset was collected from more challenging environments, such as diverse low-light indoors and outdoors scenes, making real-world noise modeling more difficult than the SIDD dataset. Hence, our future research is to bridge the gap between varied environments. We highly appreciate your insightful suggestions to steer the future direction of this field.
>
> | DLSR Model | KLD↓ | AKLD↓ |
> | --- | --- | --- |
> | Fuji | 0.0480 | 0.237 |
> | Sony | 0.1010 | 0.437 |
>
> > Reference
>
> [1\*] Chen, Chen, et al. "Learning to see in the dark." CVPR. 2018.
>
> [2\*] Zamir, Syed Waqas, et al. "Restormer: Efficient transformer for high-resolution image restoration." CVPR. 2022.

---

### Author Response · Authors · 2023-11-16
**Response to ALL reviewers**

We deeply thank all reviewers for taking their precious time to review our manuscript and give us valuable feedback. The reviewers have found our paper to be **solid with technical contribution** (YHPu, EqxC, s3bG), **effective** (YHPu, EqxC, s3bG, VvGJ), **novel** (EqxC), and **practical** (VvGJ). We further wish to highlight the major differences from NeCA and Flow-sRGB as follows:

- We tackle noise distribution based on camera configuration factors (e.g., smartphone model, ISO) using a **single** **unified model**. This is in contrast to NeCA, a state-of-the-art sRGB noise modeling network, which requires multiple independent networks, each handling a specific type of noise. Furthermore, NeCA utilizes five different losses to train models which requires carefully tuned hyperparameter of each loss coefficients, while our **NAFlow uses only a single proposed NLL loss** to model the real-world noise.
- Unlike other normalizing flow (NF) based networks such as NoiseFlow or sRGB-Flow, our model **does not rely on additional metadata** of the input image to synthesize noise at test time, using the proposed Noise-aware sampling that employs GMM to handle diverse noise distributions.

Throughout the application of our methods, our NAFlow consistently outperforms in noise quality and excels in denoising performance on benchmark datasets such as SIDD. We provide additional experimental results in the rebuttal for each reviewer.

---

### Comment · Area_Chair_dvqq · 2023-11-23
**[ICLR 2024 Reviewers’ feedback] Please read authors’ responses and give your feedback**

Dear Reviewers,

Thanks again for your strong support and contribution as an ICLR 2024 reviewer.

Please check the response and other reviewers’ comments. You are encouraged to give authors your feedback after reading their responses. Thanks again for your help!

Best,

AC

---

### Meta-Review · Area_Chair_dvqq · 2023-12-13

**Metareview:**

The proposed method works well with multiple camera models without the need for retraining, which shows the solidness of the method. The idea of encoding different noises into different Gaussian in a common latent space is convincing. Experimental results demonstrate the advantages of NAFlow compared to some previous methods. Some details could be further clarified. As one reviewer points out, the comparison with C2N is unfair. C2N is trained with only noisy images, while other methods are trained with paired images. This training setting should be pointed out in the experiments. Overall, all reviewers gave positive scores after the rebuttal.

**Justification For Why Not Higher Score:**

Some details could be further clarified. As one reviewer points out, the comparison with C2N is unfair. C2N is trained with only noisy images, while other methods are trained with paired images. This training setting should be pointed out in the experiments.

**Justification For Why Not Lower Score:**

Overall, all reviewers gave positive scores after the rebuttal.

---

### Decision · Program_Chairs · 2024-01-16

Accept (poster)